# A New Type of Etched Fiber Grating Hydrophone

**Wen-Fung Liu [1],*, Jia-Guan Li [1], Hung-Ying Chang [1], Ming-Yue Fu [2] and Chi-Fang Chen [3]**

[1] Department of Electrical Engineering, Feng-Chia University, Taichung 40724, Taiwan; smallfamiport@gmail.com (J.-G.L.); hungying.chang@gmail.com (H.-Y.C.)
[2] Center for General Education, Air Force Academy, Kaohsiung 82047, Taiwan; fumy@cc.cafa.edu.tw
[3] Department of Engineering Science and Ocean Engineering, National Taiwan University, Taipei 10216, Taiwan; chifang@ntu.edu.tw
* Correspondence: wfliu@fcu.edu.tw

**Abstract:** We propose a new type of fiber hydrophone composed of an etched fiber Bragg grating and a special packaging structure for detecting acoustic waves in the low-frequency band under water. The operating mechanism is based on the mechanical vibration of the fiber Bragg grating from the induced vibrating stress of acoustic pressure. The induced pressure of acoustic waves pushes the silicone rubber thin film, causing its vibration and then stretching the fiber Bragg grating, thus resulting in the grating wavelength shift which is overlapped with a tunable laser. The variation in the overlapped light intensity is transferred to an electrical signal by using a photodetector. From the experimental results, we can determine that the smaller the fiber diameter, the higher the sensitivity and frequency response. In order to confirm that this FBG hydrophone has the ability to work in high-frequency acoustic waves, this fiber grating hydrophone and a standard piezoelectric hydrophone are experimentally compared to in the same test conditions in the frequency range from 4 to 10 kHz. According to the experimental results, the fiber grating hydrophone has better responsivity than that of the conventional hydrophone. Due to the unique sensing structure design, this wide-band fiber hydrophone can be useful in long-term continuous monitoring of acoustic waves.

**Keywords:** fiber Bragg grating; fiber hydrophone; acoustic wave sensor; silicone diaphragm

## 1. Introduction

Fiber Bragg gratings (FBGs) [1–3] have recently received much attention for applications in fiber sensors and fiber lasers. Due to their compact size, light weight, high sensitivity, immunity to electromagnetic interference, and corrosion resistance, fiber gratings sensors have become a preferred choice over conventional electrical sensors. Nowadays, by means of different structure designs and packages, they are used in various sensing applications [4–11].

For sensing and communication systems under water, electromagnetic wave signals are difficult to transmit for measuring function due to the large attenuation caused mainly by water molecule absorption. In order to overcome this problem, a hydrophone based on acoustic waves can be utilized for measuring the object distance under water. Conventional hydrophones are based on a piezoelectric transducer generating an electric signal by sensing a pressure change, with the disadvantages of having a large size, being liable to corrosion and short circuiting by seawater, etc. Thus, for solving these problems, different kinds of fiber hydrophones are proposed. In general, there are five different kinds of fiber hydrophones: the Eisenmenger fiber hydrophone [12], Fabry Perot polymer film hydrophone [13], multilayer dielectric fiber hydrophone [14], interferometric displacement fiber hydrophone [15], and fiber grating hydrophone [16–21]. The operation mechanism of the Eisenmenger fiber hydrophone [8] is based on the detection of pressure variation to cause the refractive index mismatch between the tip of an optical fiber and water. The Fabry Perot polymer film hydrophone [13,22] is based on the interferometric detection of acoustically induced changes in the optical thicknesses of a thin film sensing structure deposited

on the tip of a single-mode fiber. The multilayer dielectric fiber hydrophone [14,15,23] is based on the interferometric detection of acoustically induced changes in the optical thickness of a thin-film multi-layer structure which consists of a series of thin dielectric films of alternating high and low indices to be sputtered onto the single-mode fiber tip. The interferometric displacement fiber hydrophone is based on detecting acoustically induced displacements of the tip of an optical fiber, in which the fiber forms one arm to be an interferometer for detecting its length change [15].

FBG-based hydrophones [16–21] have the advantages of high sensitivity, large dynamic range, flexibility, and multiplexing capability, and also have several potential applications including in building structure security monitoring, marine detecting, and medical measurements. For acoustic signal detection, Fisher et al. [16] proposed in-fiber Bragg grating for ultrasonic medical applications in 1997. The fiber Bragg grating hydrophone [17] was proposed by N. Takahashi et al. based on the intensity modulation of laser light in an FBG under the influence of sound pressure to show the linearity with the dynamic range of about 70 dB, and is operated in the range of 1 kHz to 3 MHz for the acoustic frequency. A. Cusano et al. [18] proposed an optical fiber hydrophone using polymer-coated fiber Bragg grating, in which an appropriate coating material was selected with an elastic modulus much lower than that of fiber. M. Moccia et al. [19] proposed a resonant hydrophone based on coated fiber Bragg gratings with ring-shaped polymers of different sizes and mechanical properties for measuring the acoustic frequency range from 4 to 35 kHz with greatly enhanced responsivity. B. O. Guan et al. [20] proposed a dual polarization fiber grating laser hydrophone by integrating a dual polarization fiber laser and an elastic diaphragm. By using a diaphragm, the acoustic wave pressure is transformed into the fiber axis force, acting on the laser cavity to change the fiber birefringence and the beat frequency between the two polarization lines. A. R. Karas et al. [21] proposed a passive optical fiber hydrophone array by utilizing fiber Bragg gratings which are interrogated using a single solid-state spectrometer to reduce the cost of the deployed system with lower ambient ocean acoustic noise.

In this study, we propose a new type of fiber grating hydrophone by using etched fiber Bragg gratings and a specially designed packaging structure for measuring the low-frequency band of acoustic waves. The HF-etched fiber technique serves to reduce the fiber diameter for increasing the sensing sensitivity [24,25]. We confirmed that this fiber hydrophone has the capability to measure high-frequency acoustic waves and that it performs better than conventional hydrophones in the frequency range of 4 to 10 kHz.

## 2. Sensing Principle

For this fiber grating hydrophone, the fiber Bragg grating is the key component which is fabricated by using an KrF excimer laser with the wavelength of 248 nm, combining a phase mask to create a period interference beam pattern, and then laterally exposing the core of single-mode fiber. This exposure induces the refractive index change in the fiber core to create a period index modulation along the fiber axis for forming a fiber Bragg grating. The grating wavelength $\lambda_B$ is obtained as follows:

$$\lambda_B = 2n_{eff}\Lambda \qquad (1)$$

where $n_{eff}$ is the grating effective refractive index and $\Lambda$ is the grating period. The grating reflectivity (or intensity) is dependent on the magnitude of index modulation. For increasing the fiber photosensitivity, the germanium-doped SMF-28 fiber is hydrogen-loaded in the pressure of 1500 psi for one week. According to Equation (1), the strain from external applied stress and pressure or a temperature change can cause variations in both the grating period and effective index. Therefore, the shifted grating wavelength is proportional to an applied stress along the fiber axis and a temperature variation. Thus, the operating mechanism of sensing the acoustic waves under water is based on the grating wavelength shift caused by the acoustic pressure.

The fiber sensing head is composed of a silicone rubber thin film inserted with an FBG, an acrylic ring, and a hollow cylinder, as shown in Figure 1. For the procedure of fabricating the sensing head, the FBG is firstly fixed on the groove of acrylic ring with an axial stress. The specifications of the FBG fabricated for the fiber hydrophone are 20 mm in length, FWHM bandwidth of around 0.3 nm, reflectivity of 95%, and grating center wavelength of 1554.6 nm (grating pitch of about 537.5 nm). Then, the spin coater is used to coat a silicone rubber thin film, which can transform the acoustic pressure into the fiber axial stress to cause the grating wavelength shift. Then, the silicone rubber thin film is set on the end-face of the hollow cylinder.

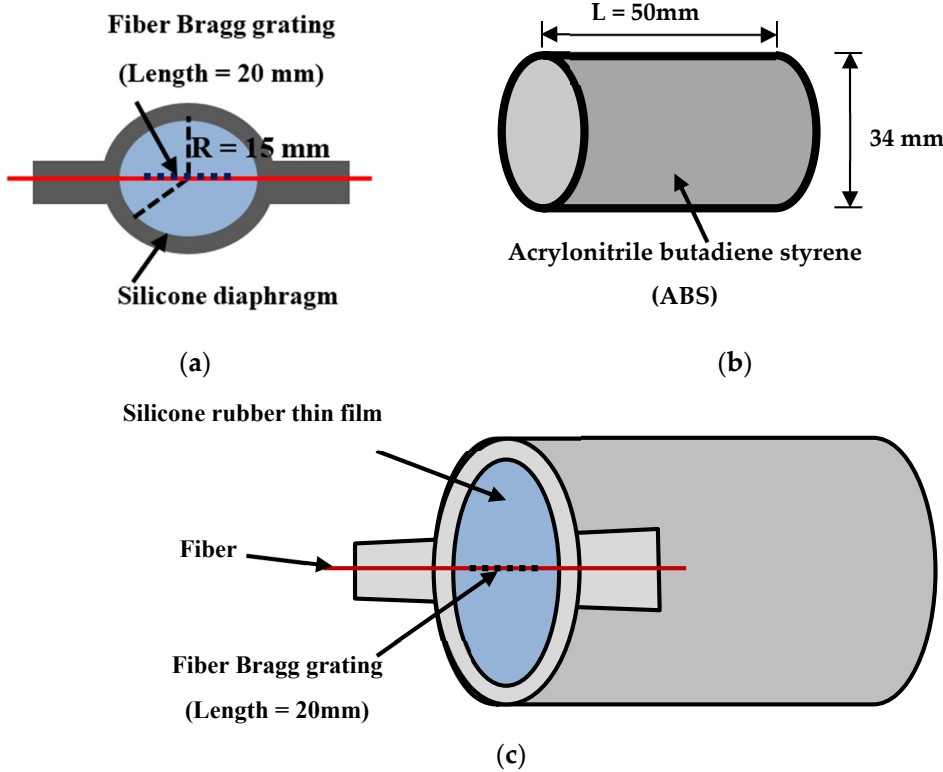

**Figure 1.** The FBG hydrophone configuration including (**a**) the silicon diaphragm, (**b**) the frame of acrylonitrile butadiene styrene, and (**c**) the FBG hydrophone composed of (**a,b**).

When acoustic pressure causes stress on the thin film, the FBG is strained to create the grating wavelength shift. The thin film is subjected to a deformation proportional to the amount of stress. The lateral strain of the film **w(r)** can be described through the following equations [8]:

$$\mathbf{w(r)} = \frac{pR^4}{64D\left[1-\left(\frac{r}{R}\right)^2\right]^2} \tag{2}$$

$$D = \frac{Et^3}{12(1-v^2)} \tag{3}$$

where **w(r)** is the transverse displacement in the deformation axis, **p** is the acoustic pressure, *R* is the radius of the thin film, *t* is the thickness of the thin film, **E** is the Young's modulus coefficient of the thin film material, and $v$ is the Poisson ratio of the thin film material.

When the acoustic pressure causes the film to produce a shape deformation, the FBG is stretched by stress $\varepsilon$, and then the grating center wavelength $\lambda_B$ is shifted. The relationship

between transverse displacement and axial strain $\mathbf{w(r)}$ from the acoustic pressure is as shown in the following formula [12]:

$$\varepsilon = \frac{48\mathbf{w(r)}y}{-2L^2} \tag{4}$$

where $\varepsilon$ is the axial stress of the fiber Bragg grating, y is the distance between the center point of deformation and the FBG, and $L$ is the film length. The grating wavelength shift $\Delta\lambda_B$ is positively correlated with the variation in both stress and temperature, which can be represented by the following formula [11]:

$$\Delta\lambda_B = \lambda_B(K_\varepsilon\varepsilon + K_T T) \tag{5}$$

where $K_\varepsilon$ and $K_T$ are pressure and temperature constants, respectively. When the temperature remains constant, the relationship between the grating wavelength shift and strain can be rewritten into:

$$\Delta\lambda_B = \lambda_B\left[\frac{48K_\varepsilon\mathbf{w(r)}y}{-2L^2}\right] \tag{6}$$

However, a good hydrophone should not only be able to clearly distinguish the frequency but also have the same sensitivity at different frequencies. Thus, the high-frequency transducer will generate a sound wave as the sound pressure level (*SPL*) in a frequency. The sound pressure level will cause hydrophones to detect voltage signals, which relates to the sensitivity of hydrophone and amplify gain. The relationship equation is as follows:

$$VL(dB) - SEN(dB) - Gain(dB) = SPL(dB) \tag{7}$$

where *SPL* is a sound pressure level, *VL* is a voltage level of measuring, *Gain* is an amplify gain, and the *VL* can be expressed with the following formulas:

$$VL = 20log_{10}Vrms \tag{8}$$

$$V_{rms} = \frac{V_{p-p}}{2\sqrt{2}} \tag{9}$$

where $V_{rms}$ is the root mean square of voltage and $V_{p-p}$ is the measured peak-to-peak voltage. With Equations (7) and (8) substituted into Equation (6), the converted relation can be described by the following equation:

$$20log\frac{V_{P-P}}{2\sqrt{2}} - SEN - Gain = SPL \tag{10}$$

Because the acoustic wave source is the same, the relationship between the standard hydrophone and the FBG hydrophone may be expressed as

$$20log\frac{V_{1P-P}}{2\sqrt{2}} - SEN_1 = 20log\frac{V_{2P-P}}{2\sqrt{2}} - SEN_2 - Gain \tag{11}$$

where $V_{1P-P}$ is the peak-to-peak voltage of the standard hydrophone, $V_{2P-P}$ is the peak-to-peak voltage of the hydrophone, $SEN_1$ is the sensitivity of the standard hydrophone obtained at the factory, and $SEN_2$ is the sensitivity of the FBG hydrophone. By detecting both the $V_{1P-P}$ and $V_{2P-P}$, the sensitivity of the FBG hydrophone can be expressed as

$$SEN_2 = SEN_1 - 20log\frac{V_{1P-P}}{2\sqrt{2}} + 20log\frac{V_{2P-P}}{2\sqrt{2}} - Gain \tag{12}$$

## 3. Experimental Setup and Results

The experimental setup to measure low-frequency acoustic waves by using the FBG hydrophane includes a signal generator, a power amplifier, a loudspeaker, an FBG hydrophone, a rectangular water tank (2 × 1 × 1 m), an optical circulator, a tunable laser, a photodetector, and an oscilloscope, as shown in Figure 2.

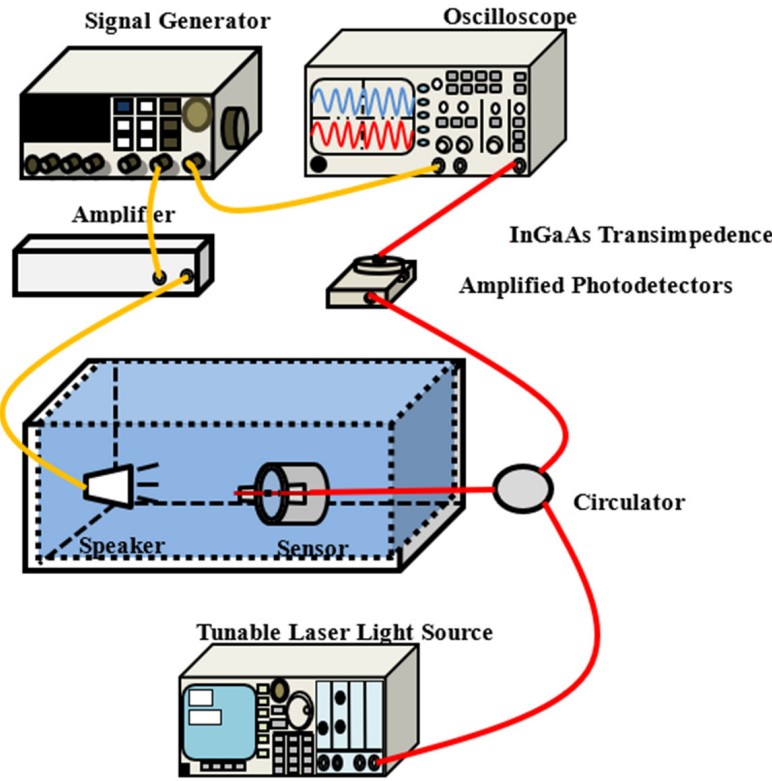

**Figure 2.** The experimental setup of using FBG hydrophone to detect low-frequency acoustic waves.

From Figure 2, we can see that the loudspeaker and the FBG hydrophone are placed inside the water tank that is filled with water. Owing to the simple insulated loudspeaker, it is difficult to generate high-frequency acoustic waves under water. Thus, acoustic waves of the low-frequency band are only generated by the loudspeaker, which is driven by a signal generator and a power amplifier. The FBG hydrophone is positioned in front of the speaker with the separation of around several centimeters. The wavelength of the tunable laser is tuned to be identical to the wavelength of the FBG hydrophone to obtain the maximum overlapped optical power, as shown in Figure 3. When the acoustic pressure is applied to cause the sensing grating wavelength shift, the variation frequency of the overlapped light power between the grating and the tunable laser will be synchronized with the acoustic wave frequency. From the experimental results, the variation frequency of overlapped light power is confirmed to be identical to the acoustic frequency. The schematic configuration of measuring acoustic frequency is illustrated in Figure 4.

The acoustic wave generated from the loudspeaker is measured by the FBG hydrophone in the format of a light signal created by the overlap between the tunable laser output power and the reflection spectrum of the sensing grating. The overlapped light power variation from the output port of the circulator is launched into the photodetector (PD) for obtaining the transferred electrical signal, which is sent to an oscilloscope for comparing the original acoustic signal. One of the two output ports of the signal generator is connected to the input of the signal amplifier to attain the larger electrical signal to drive the loudspeaker for obtaining the acoustic wave. The other output port of the signal generator is connected to the input channel 1 (above curve) of the oscilloscope to

compare the measured acoustic signal of channel 2 (below curve) from the output terminal of the photodetector to confirm that the two signals have identical frequency, as shown in Figure 4. From this figure, we can see that the signals in the two channels have the same frequency of 130.73 Hz. Moreover, by means of changing the frequency or amplitude of the signal generator, we can also see that the two channel signals in the oscilloscope are simultaneously varied but with a slight phase shift.

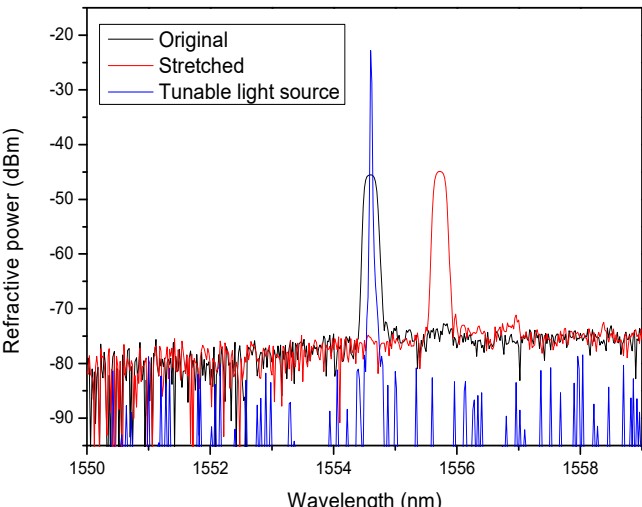

**Figure 3.** The spectra both of laser and grating wavelength shift for detecting the acoustic frequency.

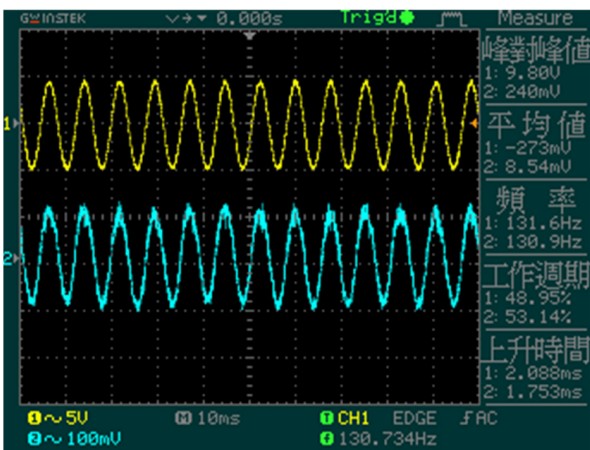

**Figure 4.** The comparison between the original electrical signal and the FBG hydrophone detecting signal shown in an oscilloscope.

By using the HF-etching fiber technique to increase the sensing sensitivity, three FBGs with different fiber diameters of 33, 71, and 97 μm are obtained for three FBG hydrophones. The experimental results of measuring the low-frequency acoustic waves are shown in Figure 5. From this figure, the fiber diameter of 33 μm has the best sensitivity, showing that the thinner the fiber diameter, the better the sensitivity and response. Acoustic signals above the frequency of 200 Hz cannot be detected, as the acoustic wave power in the frequency above 200 Hz becomes very small and difficult to measure with this FBG hydrophone, as shown in Figure 5.

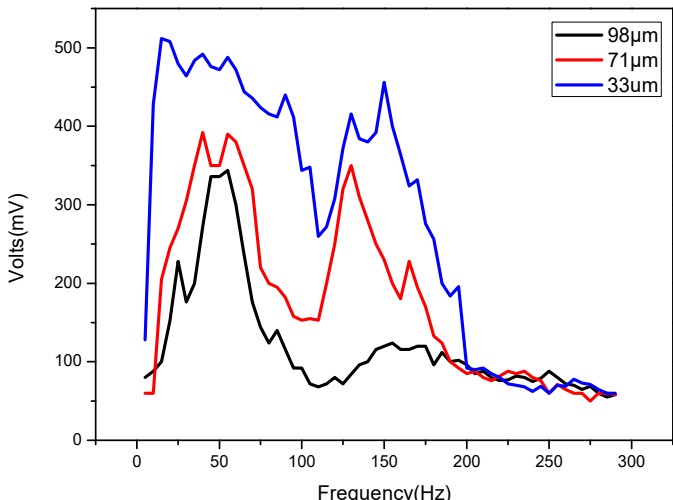

**Figure 5.** The detecting electrical signal versus different fiber diameters.

To confirm that this FBG hydrophone has the ability to detect high-frequency acoustic waves, the experimental setup used to detect acoustic waves with the FBG hydrophone, except for the signal generator, amplifier, loudspeaker, and water tank, was moved to the Underwater Acoustic Laboratory of the Department of Marine Engineering of Taiwan University to measure acoustic waves in the frequency range from 4 to 10 kHz. The Underwater Acoustic Laboratory has the most standard measurement facility for detecting acoustic waves under water, including a signal processor, an amplifier, a conventional standard hydrophone, an acoustic wave source, and a large water pool with length of 120 m, width of 8 m, and depth of 4 m, as shown in Figure 6. This conventional standard hydrophone is based on a piezoelectric transducer, transferring the detected acoustic waves to the voltage signal in the high-frequency band to be shown in an oscilloscope. The FBG hydrophone is based on the photodetector, which converts the optical signal into the electrical signal to be shown in an oscilloscope. The experimental results of both the FBG hydrophone and the standard hydrophone can be compared to determine which kind of hydrophone demonstrates better sensing performance.

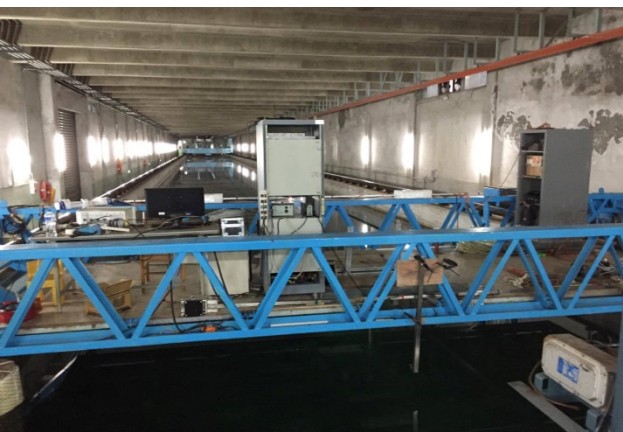

**Figure 6.** The standard measurement system of detecting acoustic waves in the Underwater Acoustic Laboratory of the Department of Marine Engineering of Taiwan University.

After the standard hydrophone is calibrated, the FBG hydrophone and standard hydrophone are located 2 m away from the acoustic wave source for receiving the same level of acoustic wave pressure. The separation between the FBG hydrophone and standard hydrophone is 1 m to avoid mutual interference, as shown Figure 7. Finally, the sensitivity

of the FBG hydrophone is calculated by measuring the sound pressure level (SPL) and peak-to-peak voltage value (VL).

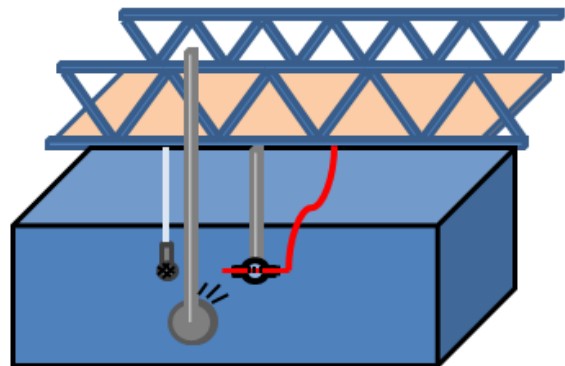

**Figure 7.** Experimental platform by using the FBG hydrophone and the standard piezoelectric hydrophone.

A frequency transducer (TR-208A) is used to generate the acoustic wave with the frequency of 4 kHz and 6 pulses per signal package. From Figure 8, the top blue signal is the original output signal from the generator to be output to drive the transducer (TR-208A), the middle black signal is the acoustic wave measured by the standard hydrophone (8103, B&K), and the bottom signal is the acoustic wave detected by the FBG hydrophone. From the comparison of these experimental results, we can see that the proposed FBG hydrophone has better sensitivity than that of the standard piezoelectric hydrophone. For detecting the original acoustic wave of 6 pulses, there are 20 pulses to be measured by the two hydrophones. This is because the echo caused by the acoustic wave is reflected back to the hydrophone.

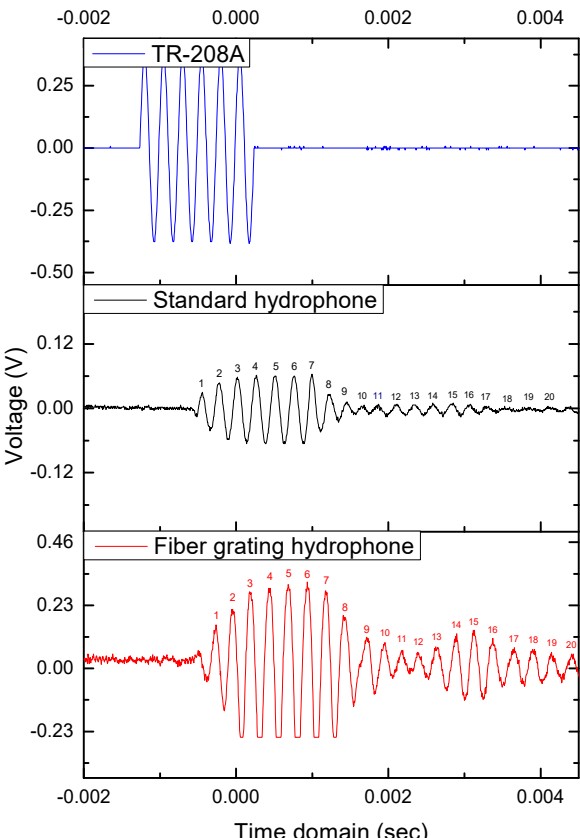

**Figure 8.** Electrical signals for a frequency of 4 kHz.

Moreover, for changing the acoustic frequency from 4 to 10 kHz with 1 kHz per step, the experimental results of using the FBG hydrophone with the fiber diameter of 25 μm are shown in Figure 9. According to this figure, the sensitivity or responsiveness of the FBG hydrophone is inversely proportional to the acoustic frequency. This is attributed to three factors, namely, the silicone diaphragm size, the FBG length, and the packaging materials. Different sensing areas and different lengths of FBG for detecting acoustic waves are similar to different lengths of guitar strings with different frequency responses. For the third factor, the hardness, elasticity, and ductility of the packaging material will affect the sensing performance. In the future, the sensing size and packaging materials will be changed to obtain better performance.

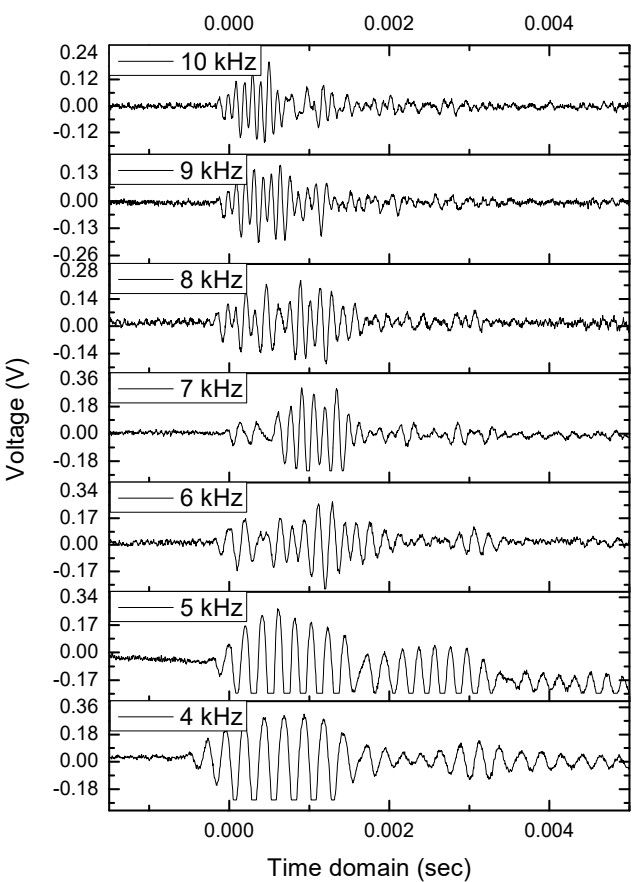

**Figure 9.** The electrical signals of a fiber Bragg grating hydrophone for different frequencies in the range of 4 to 10 kHz.

Figure 10 shows the sensitivity curves of the FBG hydrophone versus the frequency of acoustic waves with three different fiber diameters of 80, 60, and 25 μm. From this figure, we can observe that the FBG hydrophone of 25 μm has the highest sensitivity, because the smaller the fiber diameter, the easier the fiber bending. The experimental results reveal that the FBG hydrophone has a suitable performance to measure acoustic signals under water. Although the frequency response of the FBG hydrophone is not wide-band in comparison with the other types of hydrophones, the sensing head can be redesigned to improve the performance in the future.

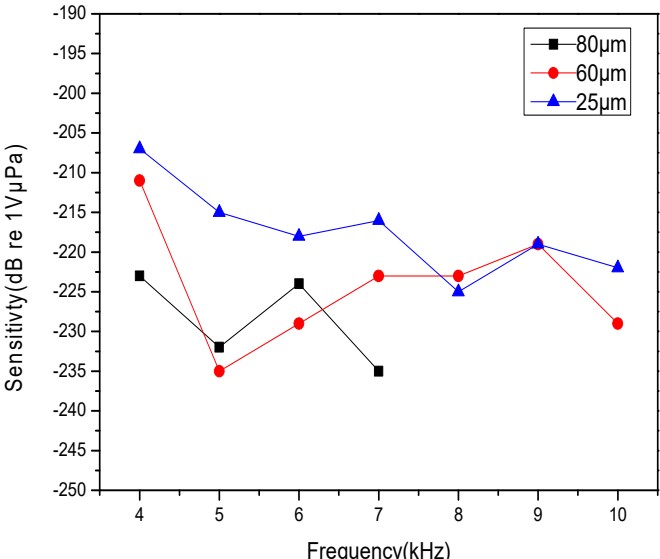

**Figure 10.** The sensitivity curves of fiber grating hydrophone versus the acoustic wave frequency in three different fiber diameters.

**4. Conclusions**

In this study, the proposed hydrophone based on FBGs is a simple and new type of fiber hydrophone. The experimental results demonstrate that the FBG hydrophone has better sensitivity than that of the commercial standard hydrophone, illustrating that the smaller the fiber diameter, the higher the sensitivity and frequency response. The frequency response and sensitivity can be improved by reducing the fiber diameter. Moreover, the performance of this FBG hydrophone can be optimized by changing several parameters, including the fiber axial stress, fiber diameter, silicone thin film thickness, and thin film radius.

**Author Contributions:** Conceptualization, W.-F.L.; methodology, W.-F.L., J.-G.L. and H.-Y.C.; formal analysis, W.-F.L. and J.-G.L.; investigation, W.-F.L., J.-G.L., H.-Y.C. and M.-Y.F.; resources W.-F.L. and C.-F.C.; data curation, W.-F.L. and J.-G.L.; writing—original draft preparation, W.-F.L., J.-G.L., H.-Y.C. and M.-Y.F.; writing—review and editing, W.-F.L., H.-Y.C., M.-Y.F. and C.-F.L.; supervision, W.-F.L.; project administration, W.-F.L.; funding acquisition, W.-F.L. All authors have read and agreed to the published version of the manuscript.

**Funding:** This research was funded by the Ministry of Science and Technology, Taiwan (contract no. MOST 110-2221-E-035-055).

**Institutional Review Board Statement:** Not applicable.

**Informed Consent Statement:** Not applicable.

**Data Availability Statement:** Not applicable.

**Acknowledgments:** The authors would like to thank the Ministry of Science and Technology, Taiwan, for sponsoring this research under contract no. MOST 108-2221-E-035-075-MY2 and no. MOST 110-2221-E-035-055-MY2.

**Conflicts of Interest:** The authors declare no conflict of interest.

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
