# Peer review of "A New Type of Etched Fiber Grating Hydrophone"

_photonics, doi:10.3390/photonics9040255_

Round 1
Reviewer 1 Report
This manuscript describes the development and test of a novel kind of FBG hydrophone. This is a very interesting work due to the viability of the sensor for practical scenarios and also because of the great importance of continuous monitoring of acoustic waves. The paper is mostly practical, deeply describing all the fabrication and implementation steps required to replicate the experiments. However, section 2 provides the theoretical model describing the sensing principle, being correct to the best of my knowledge. The results are very relevant, since they demonstrate underwater detection of acoustic waves in the facilities of the Underwater Acoustic Laboratory of the Department of Marine Engineering, being a perfect example of a practical scenario. Additionally, the paper is well written, making a proper use of the English language. It features a good Introduction, as well as relevant figures illustrating each step of the work. And most importantly, the Conclusions are perfectly aligned with the obtained results. All in all, this is a very nice piece of work and a perfect match for Photonics. I will just recommend some minor, but mandatory revisions before its consideration for publication:
- The authors mention the fabrication technique used to produce the FBGs and they mention that the FBGs have a length of 20mm. Could they provide the rest of the fabrication parameters (grating pitch, central wavelength)?
- The interrogation setup includes a TLS, which is a quite expensive element when trying to commercialize this sensor. There are ways of replacing it by a more cost-efficient approaches for the same purpose, such as the one described in IEEE Sensors Journal 13(4), 1315-1319 (2012). Could the authors mention this work and discuss if this approach could be used in this case?
- The authors mention HF-etching as a technique to increase the sensitivity of the sensors, but they do not provide any background information for non-expert readers. I recommend to mention that HF etching is a well-known technique used for the development of many kind of high-sensitive sensors, such as the one described in Biosensors and Bioelectronics 146, 111765 (2019).
- When the authors use HF-etching to increase the sensitivity of the FBGs, do they etch also the part of the optical fiber that is hold to the diaphragm or just the FBG region?
Author Response
Thanks for your comments!
The attached file is the response.

Reviewer 2 Report
This manuscript demonstrated a new type of fiber hydrophone composed of an etched fiber Bragg grating and a special packaging structure for detecting acoustic waves in the low frequency band under water. The results are rich and interesting. However, it lacks of theoretical supports.
Firstly, in the Figure 5, by using HF-etching fiber technique to increase the sensing sensitivity, three FBGs with different fiber diameters of 33, 71, and 97 micro-meters are obtained for achieving three FBG hydrophones. While, Figure 10 shows the sensitivity curves of the FBG hydrophone versus the frequency of acoustic wave in three different fiber diameters including 80, 60, and 25 micro-meters.
Moreover, 25 micro-meter is the optimal diameter for the hydrophone? Any theoretical supports?
Thirdly, the proposed hydrophone is just suitable for the acoustic frequency from 4 to 10 kHz? Why?
Author Response
Thanks for your comments!
The attached fie is the response.

Reviewer 3 Report
In this paper, the authors present an optical fiber hydrophone based on FBG and a special packaging structure for detecting acoustic waves in the low-frequency band underwater. The results indicate, the smaller the fiber diameter, the higher the sensitivity and frequency response. To confirm this FBG hydrophone with the ability to work in high-frequency acoustic waves, this fiber-grating hydrophone and a standard piezoelectric hydrophone are experimentally compared to their performance in the same test condition in the frequency range from 4k to 10k Hz. This article is clear, concise, and suitable for the scope of the journal. Several small suggestions are supplied:
1. Suggest the authors improve the title with more detail about the research.
2. Suggest the authors supply one pic of the fiber structure in Fig.1.
3. Suggest the authors enhance the introduction part with some latest review such as:
Optical fiber sensing for marine environment and marine structural health monitoring: A review Optics and Laser Technology, 2021.
Author Response

(The authors gave the same response as above.)
